# Cohort study of the mortality among patients in New York City with tuberculosis and COVID-19, March 2020 to June 2022

**Alice V. Easton** *, **Marco M Salerno**, **Lisa Trieu**, **Erica Humphrey**, **Fanta Kaba**, **Michelle Macaraig**, **Felicia Dworkin**, **Diana M. Nilsen**, **Joseph Burzynski**

Bureau of Tuberculosis Control, New York City Department of Health and Mental Hygiene, New York City, New York, United States of America

* aeaston@health.nyc.gov

## Abstract

Both tuberculosis (TB) and COVID-19 can affect the respiratory system, and early findings suggest co-occurrence of these infectious diseases can result in elevated mortality. A retrospective cohort of patients who were diagnosed with TB and COVID-19 concurrently (within 120 days) between March 2020 and June 2022 in New York City (NYC) was identified. This cohort was compared with a cohort of patients diagnosed with TB-alone during the same period in terms of demographic information, clinical characteristics, and mortality. Cox proportional hazards regression was used to compare mortality between patient cohorts. One hundred and six patients with concurrent TB/COVID-19 were identified and compared with 902 patients with TB-alone. These two cohorts of patients were largely demographically and clinically similar. However, mortality was higher among patients with concurrent TB/COVID-19 in comparison to patients with TB-alone, even after controlling for age and sex (hazard ratio 2.62, 95% Confidence Interval 1.66–4.13). Nearly one in three (22/70, 31%) patients with concurrent TB/COVID-19 aged 45 and above died during the study period. These results suggest that TB patients with concurrent COVID-19 were at high risk for mortality. It is important that, as a high-risk group, patients with TB are prioritized for resources to quickly diagnose and treat COVID-19, and provided with tools and information to protect themselves from COVID-19.

## Introduction

Since the start of the COVID-19 pandemic, public health agencies have raised concerns that COVID-19 could make TB disease worse, and vice versa. The World Health Organization stated in 2020 that "it is anticipated that people ill with both TB and COVID-19 may have worse treatment outcomes, especially if TB treatment is interrupted" [1]. In 2021, the CDC noted that "having tuberculosis can make you more likely to get severely ill from COVID-19" [2].

**Data Availability Statement:** The New York City Department of Health and Mental Hygiene surveillance data are subject to restrictions

protecting personally identifiable health information. Data on COVID cases and deaths in NYC are publicly available (https://github.com/nychealth/coronavirus-data). We do make full data available to researchers but only through a signed Data Use Agreement with the NYC DOHMH, as required by DOHMH guidelines. BTBC-Data-Team@health.nyc.gov is the non-author contact email to which requests for data can be sent.

**Funding:** The authors received no specific funding for this work.

**Competing interests:** The authors have declared that no competing interests exist.

A high case fatality rate was observed among an early cohort of patients with current or former TB and COVID-19 [3, 4]. A study found that patients with COVID-19 in the Philippines appeared to be at higher risk of death if they also had active or previous TB (risk ratio 2.17; 95% confidence interval (CI): 1.40–3.37) [5], and a study in South Africa found that TB was associated with COVID-19 mortality (adjusted hazard ratio 2.70; 95% CI: 1.81–4.04) [6].

Meta-analyses have drawn on additional smaller studies to look at the impact of COVID-19 on mortality in TB patients [7] and the impact of TB on mortality in COVID-19 patients [8]. Mortality was significantly increased in TB patients who also had COVID-19, in comparison to TB patients without COVID-19 (risk ratio = 2.10; 95% CI: 1.75–2.51) [9]. A study in California found that people diagnosed with concurrent TB and COVID-19 in 2020 had a higher age-adjusted mortality rate than those diagnosed with either disease alone [10]. Several clinical case reports have been published as well. A systematic review of published reports on patients with concomitant COVID-19 and TB found a total death rate of 22% among 83 patient histories reported from around the world [11], whereas a survey of TB/COVID-19 patients seen in 21 hospitals in Italy during the first COVID-19 wave there found that only 1/32 patients died [12]. Finally, a cohort of patients with active or previous TB and COVID-19 from 34 countries found that a number of demographic and clinical variables (age, male gender and invasive ventilation in particular) were predictive of mortality among patients [13].

### Justification

While NYC was experiencing a devastating first wave of COVID-19 cases in March-April 2020 [14], with subsequent resurgences [15], the city continued to have one of the highest rates of TB in the US [16, 17]. This manuscript reports on the demographic profile, medical risk factors and social characteristics of patients with concurrent TB and COVID-19, and assesses associations of multiple factors with death. To our knowledge, only one population-based study has been published thus far using TB and COVID-19 surveillance data in the US [10].

### Objective

To characterize people diagnosed with TB and COVID-19 in NYC in terms of 1) whether they differ from patients with TB-alone in terms of demographic and clinical variables, and 2) whether they were more likely to die during the study period when compared with patients who had either TB or COVID-19 alone.

## Methods

The demographic and clinical characteristics and mortality outcomes of retrospective cohorts of TB patients with and without COVID-19 in NYC were examined and compared.

### Data sources

Demographic and clinical information on patients with TB in NYC was drawn from the NYC TB electronic disease surveillance and case management system (Maven 5.4., Conduent Inc., Florham Park, NJ). All patients had confirmed TB disease and were reported to the CDC as a Verified Case of TB. TB is a mandatory reportable condition in the United States, which is required to be reported within 24 hours by both providers and laboratories, resulting in high reporting completeness. Demographic and clinical variables were collected by case managers during routine chart abstractions and interviews and entered into the TB surveillance and case management system. Additional information in this database is reported by laboratories and clinical providers. Information on deaths from any cause among patients with TB is collected

by case managers and confirmed by obtaining death certificates with subsequent medical review. Chart reviews for patients are routine, intense and repeated. Patients are followed from the time their disease is reported to the NYC Department of Health and Mental Hygiene (NYC DOHMH) until treatment completion, death, or loss to follow-up after repeated efforts to return a patient to care. Data is entered and updated through the course of treatment.

Data on COVID-19 diagnoses and related deaths were obtained from the NYC communicable disease surveillance system (Maven 5.5, Conduent Inc., Florham Park, NJ). The case definitions for COVID-19 and TB are those used by the Council of State and Territorial Epidemiologists [18, 19]. Data on COVID-19 cases and deaths in NYC are available publicly for download, alongside data visualizations and in-depth variable definitions [15, 20, 21].

## Patient cohorts

**TB/COVID-19 patient cohorts.** Patients diagnosed with TB between 3/1/2020 and 6/30/2022 were deterministically matched to COVID-19 surveillance data on 10/25/2022. To avoid introducing survivor bias into comparison groups, this and other cohorts were limited to patients diagnosed with TB after 3/1/2020 (the date of the first positive test for COVID-19 in NYC) [14, 22]. The match was based on demographic identifiers including name, date of birth, social security number, phone number and address. Cases of both confirmed and probable COVID-19 were included [15, 20].

This dataset was split into two cohorts of patients with TB/COVID-19: concurrent and non-concurrent. This was done because both COVID-19 and TB have been shown to result in higher morbidity and mortality immediately after diagnosis [23–25], though some patients suffer from long-term consequences of these diseases as well [26, 27]. Patients with concurrent TB/COVID-19 included those who were diagnosed with TB and COVID-19 within 120 days (regardless of which disease was diagnosed first). As a sensitivity check, some analyses were also done where concurrent TB/COVID-19 was defined as diagnosis within 90, 60 or 30 days.

**TB comparison cohort.** The "TB-alone" dataset included patients diagnosed with TB in NYC between 3/1/2020 and 6/30/2022 who had never been diagnosed with COVID-19 as of 6/30/2022. To examine whether this cohort of patients was a typical representation of TB patients prior to the pandemic, a supplementary comparison cohort was also constructed, which only included patients who were diagnosed with TB between 2016 and 2018. This pre-pandemic cohort of patients is not the primary comparison cohort used here, as patients diagnosed with TB during the pandemic may have been different from patients diagnosed with TB pre-pandemic.

**COVID-19 comparison group.** The NYC DOHMH's dataset on COVID-19 cases and deaths in NYC was used as another supplementary comparison group. The version of this dataset published on 6/30/2022 was used to mirror the end of the study period [20]. Similar to previous work from the NYC DOHMH [14], raw case and death counts were used to calculate an overall crude case fatality rate for this population. This dataset included less information than was available for the main cohorts studied here and is thus only used as a point of reference for the crude case fatality rate for NYC residents diagnosed with COVID-19 during this period.

## Assessment of patient survival

Whether each patient was alive as of 1/20/2023 (the day the dataset was pulled) was assessed for all patients. Deaths were only included if they occurred prior to 30 days after the successful completion of treatment for TB disease. A 30-day window was used to avoid excluding patients

whose treatment was administratively closed due to their death or imminent death, even though treatment might have continued if the patient had survived.

## Statistical methods

Two-tailed Fisher exact and Wilcoxon rank sum tests were used to determine statistical differences between patients with concurrent TB/COVID-19 and patients with TB-alone with a significance threshold of p = 0.05. Individual p-values are shown without mathematical correction made for multiple comparisons. Age groups were defined as 0–44, 45–64 and 65 and above.

To examine the impact of risk factors on mortality in patients with TB, patients with concurrent TB/COVID19 who died versus those who survived were compared using two-tailed Fisher exact and Wilcoxon rank sum tests.

Cox proportional hazards regression was used to assess the hazard of death past the date of TB diagnosis for patients with concurrent TB/COVID-19 patients versus TB-alone. Time-to-event was defined as the number of days between TB diagnosis and reported death date. Variables were included in the partially adjusted model only if they were either a) significantly different in the bivariate analyses comparing patients with concurrent TB/COVID-19 who died versus those who survived or comparing patients with concurrent TB/COVID-19 versus TB-alone, or b) known to be important predictors of COVID-19 mortality. Controlling for age [28, 29], sex [30, 31] and diabetes [32, 33] were prioritized because these variables are known to be associated with COVID-19 mortality and are collected in a highly standardized manner in the TB electronic disease surveillance and case management system. Variables were excluded from the model if they were colinear with other variables in the model.

Analyses were done using the R language (Version 3.5.2) [34].

## Ethics statement

The New York City Department of Health and Mental Hygiene (NYC DOHMH) Institutional Review Board determined that this project (Protocol #20–061) was exempt pursuant to 45 CFR §46.104(d)(4)(iii). This project involves only data routinely collected as part of normal program services. Consent was not obtained for this secondary research analysis as the research activities only involved information that is regulated for public health purposes. This work was supported by New York City Department of Health and Mental Hygiene Bureau of Tuberculosis Control program funds.

## Results

There were 1,141 people who were diagnosed with TB in NYC between 3/1/2020 and 6/30/2022. Among them, 902 (79%) had TB-alone and 239 (26%) were diagnosed with both TB and COVID-19. Among these 239, 106 (44%) had concurrent TB and COVID-19 and 133 (56%) had non-concurrent TB and COVID-19. Mirroring citywide trends [15], COVID-19 diagnoses among patients with TB were pronounced during the waves of COVID-19 in NYC during this period [15]; one peaked in April 2020, the next occurred between December 2020 and April 2021, and the most recent major peak in cases occurred at the start of 2022 (Fig 1).

### Comparisons between patient cohorts

The only statistically significant difference between the characteristics of patients with TB-alone and those with concurrent TB/COVID-19 was that patients with concurrent TB/COVID-19 were less likely to be male (52% versus 65%, p = 0.01, Table 1). We did not find

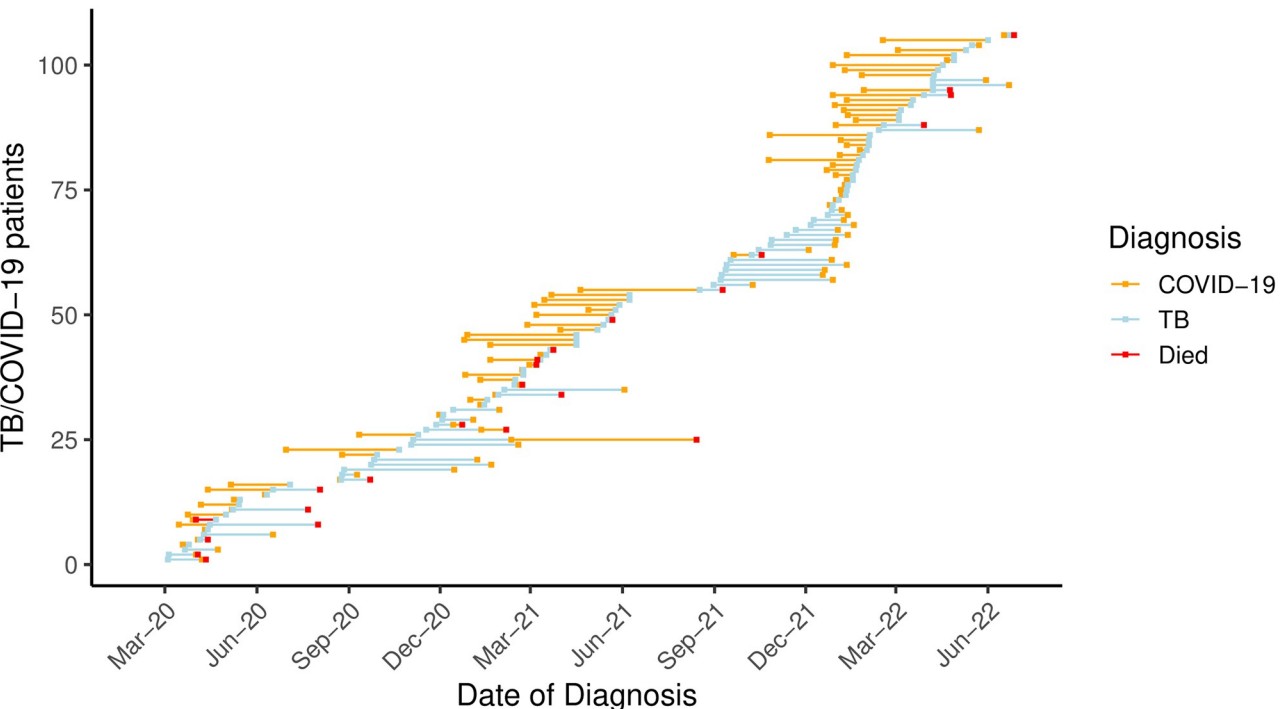

**Fig 1. Timeline of diagnoses for TB/COVID-19 patients diagnosed with TB between 3/1/2020 and 6/30/2022 (n = 106).** The dates when patients were diagnosed with TB are shown by blue points and the dates when patients were diagnosed with COVID-19 are shown by yellow points. The line between points represents the time interval between diagnosis of the two diseases. The line is the color of the first diagnosis; thus, in cases where COVID-19 was diagnosed first, the line is yellow, and in cases where TB was diagnosed first, the line is blue. Deaths are shown as red points. Only patients with concurrent TB/COVID-19 are shown (n = 106). Patients with non-concurrent TB/COVID-19 are shown in S1 Fig (n = 133). Twenty-three deaths occurred in the concurrent cohort and nine deaths occurred in the non-concurrent cohort.

significant differences between patients with concurrent TB/COVID-19 versus patients with TB-alone in terms of age, whether the patient was US-born, race/ethnicity among US-born patients, TB disease site, whether the patient ever had a cavitary chest X-ray, whether the patient was ever sputum smear positive, HIV status, social risk factors for TB, and whether the patient had MDR-TB (Table 1).

However, the percentage of patients with concurrent TB/COVID-19 who died was significantly higher than the percentage of patients with TB-alone who died (22% versus 12%, p = 0.005, Table 2). The difference in the percentage of patients who died was significant regardless of the interval between diagnosis of TB and COVID-19 considered to be "concurrent" (within 120, 90, 60 or 30 days of each other; Table 2). Among patients with concurrent TB/COVID-19 and TB-alone, similar percentages of deaths were classified as related to TB by DOHMH physicians based on death certificates (57% and 60%, p = 0.82), and similar percentages of patients were hospitalized for TB (54% and 51%, p = 0.68, Table 2).

While deaths among patients with concurrent TB/COVID-19 under the age of 45 were relatively rare (1/36, 3%), 31% (22/70) of patients aged 45 years and above died (Table 2). Among patients 65 years and above, 46% of patients with concurrent TB/COVID-19 died while 24% of patients with TB-alone died and 24% of patients with COVID-19 died (Table 2, S3 Table). The number of deaths seen here is distressingly high; this was also true for TB patients prior to the pandemic, though the overall death rate was lower pre-pandemic (8% pre-pandemic versus 12% in the TB-alone group, S2 Table). This is similar to the 8.7% of patients with TB diagnosed in NYC between 2004 and 2013 who died from any cause [35].

**Table 1. Comparison between patients diagnosed with TB-alone between 3/1/2020 and 6/30/2022 and patients with concurrent TB/COVID-19 in terms of clinical and demographic variables.**

| Characteristic | TB-alone (n = 902) | Concurrent TB/COVID-19 (n = 106) | p-value |
|---|---|---|---|
| Median [IQR] Age in Years at TB diagnosis | 51 [35, 68] | 55 [35, 64] | 0.74 |
| Male sex | 583 (65%) | 55 (52%) | 0.01* |
| US-born | 110 (12%) | 8 (8%) | 0.20 |
| Race / Ethnicity (among US-born) | | | 0.46 |
| Non-Hispanic White | 15 (14%) | 0 (0%) | |
| Non-Hispanic Black or | 58 (53%) | 4 (50%) | |
| African American | | | |
| Hispanic | 25 (23%) | 4 (50%) | |
| Asian | 9 (8%) | 0 (0%) | |
| Other / Unknown | 3 (3%) | 0 (0%) | |
| Median [IQR] years living in US+ | 12 [4, 25] | 14 [6, 28] | 0.12 |
| Pulmonary involvement | 756 (84%) | 86 (81%) | 0.49 |
| Cavitary Chest X-ray | 149/756 (20%) | 13/86 (15%) | 0.39 |
| Ever Sputum Smear positive | 421/756 (56%) | 47/86 (55%) | 0.91 |
| Multi-drug resistant (out of culture positive) | 12/758 (2%) | 2/93 (2%) | 0.66 |
| History TB disease (documented or self-reported) | 54 (6%) | 8 (8%) | 0.52 |
| Other health problems | | | |
| Diabetes | 218 (24%) | 30 (28%) | 0.34 |
| IV Status | | | 0.77 |
| Infected | 40 (4%) | 4 (4%) | |
| Uninfected | 716 (79%) | 82 (77%) | |
| Unknown/refused | 146 (16%) | 20 (19%) | |
| Social risk factors within the past 12 months prior to diagnosis | | | |
| Homelessness | 37 (4%) | 3 (3%) | 0.79 |
| Incarceration | 5 (1%) | 0 (0%) | 1.00 |
| Injection drug use | 2 (0%) | 0 (0%) | 1.00 |
| Non-injection drug use | 61 (7%) | 2 (2%) | 0.05 |
| Alcohol abuse | 24 (3%) | 1 (1%) | 0.51 |
| Smoked tobacco | 148 (16%) | 12 (11%) | 0.21 |
| Median [IQR] days from cough onset to TB diagnosis ++ | 50 [20, 93] | 62 [22, 94] | 0.71 |
| Ever on Directly Observed Therapy for TB (among eligible) | 570/761 (75%) | 64/82 (78%) | 0.59 |

+Among individuals who were born outside the US, where this information is available. Based on 792 and 98 individuals, respectively.

++Among individuals for whom a date of cough onset is available. Based on 495 and 127 individuals, respectively.

Fisher Exact tests and Wilcoxon rank sum tests were used to calculate p-values.

Significance levels:

* p < 0.05,

** p <0.01,

*** p<0.001.

## Concurrent TB/COVID-19 patients who died vs those who survived

A comparison of the 23 patients with concurrent TB/COVID-19 who died with the 83 who survived found that patients who died were significantly older (median age 65 versus 48, p<0.001) and more likely to have diabetes (52% vs 22%, p = 0.008). No other significant differences were identified between patients with concurrent TB/COVID-19 who died versus those who survived (Table 3). The sample size was too limited to allow for exploration of additional

**Table 2. Comparison between patients diagnosed with TB-alone between 3/1/2020 and 6/30/2022 and patients with concurrent TB/COVID-19 in terms of outcomes.**

| Characteristic | | TB-alone (n = 902) | Concurrent TB/COVID-19 (n = 106) | p-value |
|---|---|---|---|---|
| Hospitalized for TB | | 464 (51%) | 57 (54%) | 0.68 |
| Deaths | | 105 (12%) | 23 (22%) | 0.005** |
| Deaths, stratified by the interval between TB and COVID-19 diagnoses + | Within 90 days | " | 20/84 (24%) | 0.003** |
| | Within 60 days | " | 18/66 (27%) | <0.001*** |
| | Within 30 days | " | 13/42 (31%) | 0.001** |
| Deaths prior to treatment initiation | | 26/105 (25%) | 9/23 (39%) | 0.20 |
| Deaths, stratified by age | 0 to 44 | 16/377 (4%) | 1/36 (3%) | 1.00 |
| | 45 to 64 | 24/253 (9%) | 10/44 (23%) | 0.02* |
| | 65+ | 65/272 (24%) | 12/26 (46%) | 0.02* |
| Death was related to TB | | 63/105 (60%) | 13/23 (57%) | 0.82 |

+ As a sensitivity check on whether the mortality comparisons are affected by the definition of concurrent TB/COVID-19 as within 120 days, analyses were also done where concurrent TB/COVID-19 was defined by diagnoses within shorter periods of days. All comparisons are with the TB-alone group (n = 902).

Fisher Exact tests were used to calculate p-values.

Significance levels:

* p < 0.05,

** p <0.01,

*** p<0.001.

TB risk factors; for example, only two of the 106 patients with concurrent TB/COVID-19 had MDR-TB. The association between age, diabetes and death shows the importance of considering these variables in regression analyses, discussed below.

Multivariable Cox proportional hazards regression suggests that the hazard of death is 2.62 (CI: 1.66–4.13, p<0.001) times higher for patients with concurrent TB/COVID-19 when compared to patients with TB-alone after controlling for age and sex. Since age and diabetes are

**Table 3. Comparison of concurrent TB/COVID-19 patients who died versus those who survived.**

| Characteristic | Died (n = 23) | Survived (n = 83) | p-value |
|---|---|---|---|
| Median [IQR] Age in Years at TB diagnosis | 65 [57, 73] | 48 [30, 60] | <0.001*** |
| Male | 13 (57%) | 42 (51%) | 0.65 |
| US-born | 3 (13%) | 5 (6%) | 0.37 |
| Diabetes | 12 (52%) | 18 (22%) | 0.008** |
| Pulmonary involvement | 17 (74%) | 69 (83%) | 0.37 |
| Cavitary Chest X-ray | 2/17 (12%) | 11/69 (16%) | 1.00 |
| Ever Sputum Smear positive | 7/17 (41%) | 40/69 (58%) | 0.28 |
| Hospitalized for TB | 11 (49%) | 46 (55%) | 0.64 |
| Ever on DOT (among eligible for DOT) | 3/5 (60%) | 61/77 (79%) | 0.30 |
| MDR (among culture positive) | 1/21 (5%) | 1/72 (1%) | 0.39 |

Fisher Exact tests and Wilcoxon rank sum tests were used to calculate p-values.

Significance levels:

* p < 0.05,

** p <0.01,

*** p<0.001.

correlated ($R^2 = 0.12$, p<0.001 for the 1,141 patients diagnosed with TB between 3/1/2020 and 6/30/2022), this model was not adjusted for diabetes. As a sensitivity check, when patients are subsetted into those with and without diabetes, the results of this regression were similar (S1 File). No other variables were significantly different in the bivariate analyses comparing patients with concurrent TB/COVID-19 who died versus those who survived (Table 3) or between patients with TB-alone versus concurrent TB/COVID-19 (Table 1), so no additional variables were included in this partially adjusted regression model. The unadjusted hazard of death was similar at 2.18 (CI: 1.39–3.42, p<0.001). These regressions were based on 1008 individuals (902 with TB-alone and 106 with concurrent TB/COVID-19) and 128 events (105 deaths among patients with TB-alone and 23 among patients with concurrent TB/COVID-19).

## Mortality among supplementary comparison groups

When compared with patients with TB-alone, a similar percentage of patients with non-concurrent TB/COVID-19 died (S1 Table). However, a lower percentage of patients with COVID-19 alone and TB diagnosed pre-pandemic died (S2 and S3 Tables).

## Discussion

NYC was an early epicenter of COVID-19 in the US [14, 36]. With NYC having one of the highest rates of TB in the country, the convergence of these two respiratory diseases (in addition to reduced public health services due to lockdown measures) raised concerns for TB control staff of downstream impact, including mortality [16]. Deaths from COVID-19 peaked in early April and then declined rapidly [14]. However, during routine reviews of patients with TB, it appeared that the number of patients who were dying was particularly high in April of 2020 and subsequently, particularly high among patients who had COVID-19 as well. This work aimed to determine whether this observation was borne out in the data and investigate what could be done to improve outcomes for future patients with both TB and COVID-19.

### Comparing characteristics of TB/COVID-19 versus TB-alone patients

One possible explanation for why NYC observed an increase in deaths among patients with both TB and COVID-19 was that these patients were at high risk for mortality due to their demographic or clinical characteristics. An early multinational study found that while the mortality rate was high among patients with both TB and COVID-19, this could have been the result of older age and other comorbidities in the study population [3, 4]. In California, researchers found differences in terms of diabetes prevalence and ethnicity between patients with concurrent TB/COVID-19 and a comparison group of patients who had active TB prior to the COVID-19 pandemic [10].

Surprisingly, the TB-alone and concurrent TB/COVID-19 patient cohorts in NYC were similar to each other in terms of a range of demographic and clinical variables. Since these two cohorts were similar across most available variables, our comparisons of mortality among patients with TB-alone and patients with concurrent TB/COVID-19 are less likely to be biased by fundamental differences in these two populations. Thus, the observation that many patients with concurrent TB/COVID-19 were dying could not be explained simply by observable differences between patients with TB-alone versus patients with concurrent TB/COVID-19.

Even after doing sensitivity checks on how we defined "concurrent" TB/COVID-19 (as diagnoses within 120, 90, 60, or 30 days) and controlling for patient age and sex, death rates were significantly higher among patients with concurrent TB/COVID-19 than among patients with TB-alone. It is surprising that the percentage of patients with concurrent TB/COVID-19 who died was so high, given that the period covered included not only the "lockdown" period

in NYC, but also 18 months in 2021 and 2022 in which vaccination and access to anti-viral and other treatments became increasingly widespread.

When comparing patients with concurrent TB/COVID-19 who died versus those who survived, the only significant differences identified were that patients who died were more likely to be older and more likely to have diabetes. In previous work, it was found that among both US-born and non-US-born patients with TB, TB-related death was associated with increased age, being culture positive and having both pulmonary and extra-pulmonary disease, though additional demographic and social risk factors were associated with TB-related death for non-US-born patients specifically [35].

Finally, we investigated whether the high death rates observed in patients with concurrent TB/COVID-19 were similar to death rates among patients with COVID-19 of similar ages. However, we found that death rates among patients in NYC with COVID-19 alone were not as high as death rates among patients of similar ages with concurrent TB/COVID-19.

## Limitations

A relatively small sample size and limited period of follow-up time prevented us from thoroughly examining the impact of a wider range of potential confounding demographic and clinical variables on patient outcomes. Furthermore, the data available for each patient was limited to what is routinely collected for patients with TB in NYC, and thus did not include information on the severity of COVID-19 disease, or the treatments received during an episode of COVID-19.

While the availability of disease and mortality data for both TB and COVID-19 is a strength of this study, it is likely that many cases of COVID-19 went undiagnosed in NYC during the study period—including in some patients with TB. Early on, access to tests for COVID-19 was severely limited, leading to test positivity rates that peaked at 69% on 3/30/2020 [15]. Later, the use of at-home tests likely led to a rise in unreported COVID-19 cases [37]. While many hospital-based clinicians ordered COVID-19 tests for patients during their workup for TB, and clinicians in TB clinics often recommended getting COVID-19 tests to patients experiencing symptoms, COVID-19 was not systematically ruled out as part of the care and monitoring of the patients with TB included in this study.

TB data quality is high due to intensive case management, but the match between the TB and COVID-19 datasets is also reliant on accurately reported demographics in the COVID-19 database. This depends on accurate COVID-19 test reporting by laboratories. Missed matches could lead to an underestimate of the number of patients with TB/COVID-19.

If any of the deaths that occurred in patients in the TB-alone cohort were due to undiagnosed COVID-19 around the time of their death, or to COVID-19 laboratory reporting data quality issues, this could have artificially reduced our measurement of the impact of COVID-19 on mortality in patients with TB.

## Future work

The level of mortality that results from concurrent TB/COVID-19 in the future will depend both on risk mitigation tools available to each patient (including vaccination and treatment options) as well as the risk profile of the SARS-CoV-2 variant that has infected each patient.

We will continue to see additional patients with concurrent TB/COVID-19. It will remain important to critically assess short and long-term outcomes for these patients. Future studies from broader geographies and covering a longer period will enable the examination of case fatality rates for patients with TB/COVID-19 over time and in comparison with other patient populations. Future work could also examine the mortality of patients who are diagnosed with

COVID-19 at any time when they are being treated for TB, or after they have been cured of TB. As COVID-19 surveillance is incorporated into standard practices in health departments around the US, it will hopefully become easier to integrate datasets from different agencies and jurisdictions.

## Conclusions

We found that patients in NYC with concurrent TB/COVID-19 were at high risk for mortality, particularly if they were 45 years or older. The difference between the frequency of death among patients with concurrent TB/COVID-19 and TB-alone could not be explained by differences in demographic or clinical characteristics between these groups. Our results are aligned with findings of elevated mortality risk among patients with TB/COVID-19 in California, Asia and South Africa [5–8, 10], and support statements of concern about outcomes for patients with both TB and COVID-19 [2].

Given the growing recognition of the risks posed by these diseases in combination, it will be important to increase screening for COVID-19 among patients with TB and increase screening for TB among patients hospitalized with suspected COVID-19. Early detection of COVID-19 in patients with TB should then be followed by guidance towards easy and immediate access to anti-viral medications, and any other effective medications for COVID-19 that are developed in the future. In DOHMH TB clinics in NYC, this means that providers can refer patients with COVID-19 symptoms to co-located COVID-19 testing sites, and staff regularly refer patients with TB to sites where they can be vaccinated against COVID. Treating clinicians will also need to have access to information regarding interactions between anti-TB medications and COVID-19 treatments. These findings highlight the importance of making sure that TB is recognized as an important comorbidity for patients with COVID-19, and vice versa. Wide recognition among clinicians, public health agencies and TB patients of the risks posed to patients diagnosed with concurrent TB and COVID-19 could help ensure that the risks are mitigated for these patients.

## Supporting information

**S1 Fig. Timeline of diagnoses for non-concurrent TB/COVID-19 patients diagnosed with TB between 3/1/2020 and 6/30/2022 (n = 133).** The dates when patients were diagnosed with TB are shown by blue points and the dates when patients were diagnosed with COVID-19 are shown by yellow points. The line between points represents the time between diagnosis of the two diseases. The line is the color of the first diagnosis. Deaths are shown as red points. Nine deaths occurred among these patients; however, one death occurred after 6/30/2022, but before that patient completed treatment for TB, and is thus not visible here. Five deaths occurred during a COVID-19 wave in late 2021. Four of these deaths occurred in a hospital, and COVID-19 was not noted in their death certificates. The fifth person who died was diagnosed with COVID-19 soon after their death.
(TIF)

**S1 File. Full results of Cox proportional hazards regression analyses.**
(DOCX)

**S1 Table. Comparison of mortality among patients diagnosed with TB in NYC between 3/1/2020 and 6/30/2022 (TB-alone group) versus patients diagnosed with COVID-19 during the same period (COVID-19 alone group).**
(DOCX)

**S2 Table. Comparison of patients diagnosed with TB in NYC in 2016–2018 (pre-pandemic TB) versus those diagnosed with TB in NYC between 3/1/2020 and 6/30/2022 (TB-alone group).**
(DOCX)

**S3 Table. Comparison of patients diagnosed with TB in NYC between 3/1/2020 and 6/30/2022 and not diagnosed with COVID-19 during that period (TB-alone group), versus patients diagnosed with both TB and COVID-19 during this period, but where the TB and COVID-19 diagnoses were over 120 days apart (non-concurrent).**
(DOCX)

## Acknowledgments

This work was supported by the NYC DOHMH, New York, NY, USA. The authors thank the physicians, nurses, public health advisors, and other staff of the New York City Department of Health Chest Centers for their commitment and dedication in delivering the highest quality care to patients. We would like to thank Vasudha Reddy and her colleagues in the Bureau of Communicable Diseases at the New York City Department of Health and Mental Hygiene for sharing data used in the match between the TB and COVID-19 registries.

## Author Contributions

**Conceptualization:** Alice V. Easton, Marco M Salerno, Lisa Trieu, Erica Humphrey, Michelle Macaraig, Diana M. Nilsen, Joseph Burzynski.

**Data curation:** Lisa Trieu.

**Formal analysis:** Alice V. Easton.

**Methodology:** Alice V. Easton, Marco M Salerno, Lisa Trieu.

**Resources:** Lisa Trieu, Diana M. Nilsen, Joseph Burzynski.

**Supervision:** Michelle Macaraig, Felicia Dworkin, Diana M. Nilsen, Joseph Burzynski.

**Visualization:** Alice V. Easton.

**Writing – original draft:** Alice V. Easton.

**Writing – review & editing:** Alice V. Easton, Marco M Salerno, Lisa Trieu, Erica Humphrey, Fanta Kaba, Michelle Macaraig, Felicia Dworkin, Diana M. Nilsen, Joseph Burzynski.

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
