## [Decision Letter · Decision Letter 0]

13 Jan 2023

PGPH-D-22-01834

Mortality among patients in New York City with Tuberculosis and COVID-19

Dear Dr. Easton,

Thank you for submitting your manuscript to PLOS Global Public Health. After careful consideration, we feel that it has merit but does not fully meet PLOS Global Public Health’s publication criteria as it currently stands. Therefore, we invite you to submit a revised version of the manuscript that addresses the points raised during the review process.

We look forward to receiving your revised manuscript.

Kind regards,

Javier H Eslava-Schmalbach, M.D., Ph.D., MSc

Academic Editor

Journal Requirements:

1. Please insert an Ethics Statement at the beginning of your Methods section, under a subheading 'Ethics Statement'. It must include:

1) The name(s) of the Institutional Review Board(s) or Ethics Committee(s)

2) The approval number(s), or a statement that approval was granted by the named board(s) 

3) (for human participants/donors) - A statement that formal consent was obtained (must state whether verbal/written) OR the reason consent was not obtained (e.g. anonymity). NOTE: If child participants, the statement must declare that formal consent was obtained from the parent/guardian.

2. Please provide separate figure files in .tif or .eps format only and remove any figures embedded in your manuscript file. Please also ensure that all files are under our size limit of 10MB.

3. We notice that your supplementary material are included in the manuscript file. Please remove them and upload them with the file type 'Supporting Information'. Please ensure that each Supporting Information file has a legend listed in the manuscript after the references list.

Additional Editor Comments (if provided):

Dear authors:

We have received the reviewers' comments. Additionally to them, consider these others:

1. Add the design to the title (STROBE guidelines), and to the methods section too. 

2. The COVID-19 comparison group is not included in the objective of the study. You should clarify this. The same happens with comparisons performed with pre-pandemic patients. These patients are not part of the study period, either.

3. Calls to the figures must be made from the results section and not from the methods section. Please delete these calls in methods section.

4. Use the format of table 2 to redo table 1. Include statistics in all of the tables.

5. Statistics used should be mentioned at tables footer (not in the columns titles). They should be detailed for each variable.

6. It is note clear why didn't you control other confounders in the multivariable model. This should be cleared.

7. Table 4 should show results of the crude and adjusted analysis for all of the variables included in the multivariable analysis.

8. Double check the use of the word "multivariate" instead of "multivariable".

9. Include in the methods section the description of the time-to-event variable used to perform the survival multivariable analysis.

10. The full name of NYC DOHMH should be mentioned at Ethics section.

11. The meaning of dots and lines is not clear in Figure 1. There are more combinations of "colors and lines" than are named in the legends. This should be reviewed

12. Figure 2 is redundant, given that the same information is at Table 1. Consider to delete it. 

13. Consider to delete "Investigation of supplementary comparison populations" at supplementary section, given that is not related with the objective of the study

14. The use of third person pronouns is preferred ("it" instead of "we" or "I")

Please include/answer each one of all these comments.

Reviewers' comments:

Reviewer's Responses to Questions

**Comments to the Author**

1. Does this manuscript meet PLOS Global Public Health’s publication criteria? Is the manuscript technically sound, and do the data support the conclusions? The manuscript must describe methodologically and ethically rigorous research with conclusions that are appropriately drawn based on the data presented.

Reviewer #1: Yes

Reviewer #2: Partly

2. Has the statistical analysis been performed appropriately and rigorously?

Reviewer #1: Yes

Reviewer #2: No

3. Have the authors made all data underlying the findings in their manuscript fully available (please refer to the Data Availability Statement at the start of the manuscript PDF file)?

Reviewer #1: Yes

Reviewer #2: Yes

4. Is the manuscript presented in an intelligible fashion and written in standard English?

Reviewer #1: Yes

Reviewer #2: Yes

5. Review Comments to the Author

Reviewer #1: Excellent and much-needed analysis of mortality among people infected with both Covid and TB.

The authors have clearly listed the limitations, including the small number of deaths, inadequate adjustment of confounding, and undiagnosed Covid during the study period.

I have a few suggestions for improvement:

1. Since the study period is mostly during the first year of the pandemic, and before widespread access to vaccination, it might be good to explicitly mention this in the title (e.g. Mortality among patients in New York City with Tuberculosis and COVID-19 during the first year of the pandemic, or something like that).

2. The study validity rests a lot on the two databases: NYC TB electronic disease surveillance and case management system, and the communicable disease surveillance system in NYC. The authors need to do a better job of explaining these systems and the quality of data they provide. How complete is the reporting in these databases, and how easy is it to cross-match or do the records linkage? What unique ID is used to do the linkage and in what % of the cases was it difficult to link? What is the duration of follow-up of patients in the databases? Are there any published studies on the quality of data in these two systems?

3. The hazard ratios provided must be viewed as partially adjusted, as a full adjustment of potential confounders could not be done. So, please explicitly state this.

4. It will be good for the authors to expand the time period and do a larger study by included patients during 2021 and 2022, acknowledging the challenges in accounting for vaccination rates and testing rates that may have changed over time. Access to paxlovid might be another confounding variable. What should others who plan similar studies in future worry about in terms of design and analysis? Kindly provide some suggestions for additional research.

5. I agree with the recommendation about giving priority to people with TB for early access to Covid vaccination and would also add that they should be prioritized for easy access to anti-virals like paxlovid.

6. During the study period, how easy was it for New Yorkers to access Covid testing? what was the testing rate and positivity rate in NYC? Can you please cite data on this? If it was very difficult, then it would likely mean that Covid was under-diagnosed in the entire population (which also include those with TB).

Reviewer #2: The manuscript aims to describe the mortality amongst TB/COVID-19 patients in comparison with patients with TB only (or COVID-19 only) during the period March 2020-May 2021 in NYC. The topic is interesting, although already addressed by similar papers in different settings. However, presented data do not add much to the available knowledge and would benefit from major re-writing plus addition of details before being considered for publication. Specific comments are provided here below for authors' consideration.

1. Abstract is confusing as it contains results under method section (e.g. "We identified 44 patients who were diagnosed with TB and COVID-19..... compared to those diagnosed with TB-alone during the same period (n=421)". Please edit the method and results section of the abstract accordingly.

2. How were COVID-19 and TB diagnosis formulated' Please specify (Authors mention that both confirmed and probable COVID-19 patients were included but not details are provided). How many were confirmed TB cases? How many P/EP? Radiological findings (i.e. how many had extensive CXR abnormalities)? Some data are included in tables but not mentioned in the text. Please add as they may play a role in determining case severity and, hence, risk of death.

3. How many patients died during intensive phase of TB-treatment and how many during continuation phase? Pleas add.

4. In TB-alone patients, was COVID-19 ruled out? If so, how?

5. How many patients were hospitalized? How many in ICU?

6. How many patients did need oxigen-therapy? How many needed high-flow ventilation? Please disaggregate the mortality among different groups

7. Which treatment did COVID-19 patients receive? Anti-viral? Immunusuppressant? Steroids? Please specify.

8. Ref. 13 should be corrected as authors go under the name: "TB/COVID-19 Global Study Group"

9. Statistical analysis should be included in table 1 (and mentioned into the text). Table 2 should be the first table presented as it describes the study population characteristics.

10. Mortality is on the high side not only for TB/COVID19 patients but also for TB-alone patients (reaching 9% of total patients, with 21% in the 65+ age band in the study period and 8% in pre-pandemic time). Can authors comment on this aspect which is really worrying and propose corrective actions (i.e. COVID-19 pandemic does not seem to be an explanation for the high number of death, according to supplementary material, so there should be other factors to explain this, please elaborate on this point)? Deaths after completion of TB treatment should be counted as related to TB (if the treatment was successfully completed - so please correct 144 deaths of pre-pandemic period to 138)

11. Last but not least, it would be highly relevant to run similar analysis (i.e. comparison betwen mortality in TB/COVID-19 and TB-alone patients) including recent data, to understand if the recent approach to COVID-19, including vaccination, played a role in reducing mortality in TB/COVID-19 patients compared to the first phase of the pandemic. This last point would add very interesting information that would position this paper as a priority research.

6. PLOS authors have the option to publish the peer review history of their article (what does this mean?). If published, this will include your full peer review and any attached files.

**Do you want your identity to be public for this peer review?** For information about this choice, including consent withdrawal, please see our Privacy Policy.

Reviewer #1: No

Reviewer #2: No

---

## [Editor Report · Decision Letter 1]

13 Mar 2023

Cohort study of the mortality among patients in New York City with Tuberculosis and COVID-19, March 2020 to June 2022

PGPH-D-22-01834R1

Dear Dr. Easton,

We are pleased to inform you that your manuscript 'Cohort study of the mortality among patients in New York City with Tuberculosis and COVID-19, March 2020 to June 2022' has been provisionally accepted for publication in PLOS Global Public Health.

Best regards,

Javier H Eslava-Schmalbach, M.D., Ph.D., MSc

Academic Editor